# Are BDNF and Stress Levels Related to Antidepressant Response?

**DOI:** 10.3390/ijms251910373

**Published:** 2024-09-26

**Authors:** Mónica Flores-Ramos, Andrés Vega-Rosas, Nadia Palomera-Garfias, Ricardo Saracco-Alvarez, Gerardo Bernabé Ramírez-Rodríguez

**Affiliations:** 1Laboratorio de Epidemiología Clínica, Subdirección de Investigaciones Clínicas, Instituto Nacional de Psiquiatría Ramón de la Fuente Muñiz, Calzada México Xochimilco #101, Col. San Lorenzo Huipulco, Tlalpan, Mexico City C.P. 14370, Mexico; 2Laboratorio de Neurogénesis, Subdirección de Investigaciones Clínicas, Instituto Nacional de Psiquiatría Ramón de la Fuente Muñiz, Calzada México-Xochimilco #101, Col. San Lorenzo Huipulco, Tlalpan, Mexico City C.P. 14370, Mexico; dr.andresvegar@gmail.com (A.V.-R.); gbernabe@inprf.gob.mx (G.B.R.-R.); 3Servicio Social, Escuela Superior de Medicina, Instituto Politécnico Nacional, Salvador Díaz Mirón esq. Plan de San Luis S/N, Miguel Hidalgo, Casco de Santo Tomas, Mexico City C.P. 11340, Mexico; nadiapalomerag@gmail.com; 4Subdirección de Investigación Clínica, Instituto Nacional de Psiquiatría Ramón de la Fuente Muñiz, Calzada México-Xochimilco #101, Col. San Lorenzo Huipulco, Tlalpan, Mexico City, C.P. 14370, Mexico; dr_saracco@yahoo.com.mx

**Keywords:** BDNF, antidepressant treatment, state anxiety

## Abstract

Antidepressant response is a multifactorial process related to biological and environmental factors, where brain-derived neurotrophic factor (BDNF) may play an important role in modulating depressive and anxious symptoms. We aimed to analyze how BDNF impacts antidepressant response, considering the levels of anxiety. Methods: A total of 40 depressed adults were included. We evaluated initial serum BDNF, anxiety through the State–Trait Anxiety Inventory (STAI), and the severity of depressive symptoms by the Hamilton Depression Rating Scale (HDRS). Participants received antidepressant treatment for 8 weeks, and response to treatment was evaluated according to the final HDRS scores. Results: Basal BDNF was higher in responders compared to non-responder depressed patients, in addition to being inversely associated with the severity of anxiety and depression. Conclusions: Baseline BDNF serum is an adequate predictive factor for response to antidepressant treatment with SSRI, with lower pre-treatment levels of BDNF associated with higher anxiety symptoms after treatment. Stress levels could influence the response to treatment, but its association was not conclusive.

## 1. Introduction

Depressive disorder is a common mental health condition, affecting about 5% of adults around the world [1]. A high global burden is estimated to be derived from depression, which is associated with anxiety, suicidal behavior, and loss of well-being. Currently, many pharmacological options are available to treat depression; despite their efficacy, the reported response rate is 47%, according to the Sequenced Treatment Alternatives to Relieve Depression (STAR*D) study [2].

One of the main substrates related to major depressive disorder (MDD) and its response to treatment is neuroplasticity, which is predominantly modulated by brain-derived neurotrophic factor (BDNF) [3]. Alterations in its metabolism have been strongly associated with psychiatric, neurological, and cardiovascular disorders [4,5,6]. Specifically, during stress and MDD, a decrease in BDNF levels has been associated with neuronal atrophy, neuronal survival, and proinflammatory profiles [7]. Additionally, BDNF plays a critical role in the clinical presentation of depression, as both peripheral and central BDNF levels are lower during depressive episodes. An increase in blood BDNF levels after antidepressant treatment is observed, which is proportional to symptom improvement [8]; however, the literature on this topic remains controversial [9,10]. Moreover, the chronicity of depression [11] and the duration of the last depressive episode also appear to influence the levels of BDNF [12].

On the other hand, anxiety could modulate the antidepressant response. According to preclinical studies, chronic stress is related to depressive-like behavior [13], and animals showing anxious-like behavior had a lower antidepressant response to fluoxetine [14]. Similarly, clinical studies have demonstrated that MDD remission is significantly less likely and takes more time in patients with anxious depression [15,16]. Despite the high comorbidity between anxiety and depression, anxiety symptoms are not always considered in the evaluation of the response to antidepressant treatment.

As occurs in depression, BDNF levels may be involved in anxiety. Single prolonged stress produces a significant reduction in BDNF levels in the cerebrospinal fluid of rats, and different routes and doses of BDNF administration produce anxiolytic effects [17]. In nondepressed patients with anxiety disorder, serum BDNF levels did not differ from those of healthy controls, but an association with gender was observed: BDNF levels were lower in female patients than in female controls [18]. An increase in BDNF levels was observed in patients with generalized anxiety disorder after 15 weeks of treatment with duloxetine [19]. While in patients with panic disorder who had a poor response to cognitive behavioral therapy, significantly lower levels of serum BDNF were observed [20].

Given the crucial functions of BDNF in the diagnosis and response to the treatment of depression, it is important to understand the role it plays when depression is accompanied by anxiety symptoms, as occurs in a great percentage of MDD cases. Therefore, the current study was conducted to determine if plasma BDNF levels differ between depressed patients who respond to antidepressant treatment and those who do not respond, as well as the impact of anxiety in this condition.

## 2. Results

A total of 63 patients met the inclusion criteria for this study and were enrolled. Of them, 40 completed the eight weeks of treatment. Thirteen patients did not attend the following visits, eight discontinued the medication, and two did not complete the evaluations. According to the Hamilton Depression Rating Scale (HDRS) scores, 21 patients responded to the treatment at the final evaluation. Sociodemographic characteristics were similar in responders and non-responders (Table 1).

The clinical characteristics of participants are shown in Table 2. No significant differences were observed between responders and non-responders at the basal line in the depression and anxiety scores. Similarly, the time since the depressive symptoms started was similar between groups. It is important to mention that 57% of the patients who responded to the treatment had between zero and two previous depressive episodes, while in the non-responder group, only 21% had up to two previous depressive episodes. No differences were observed in the antidepressants prescribed between responder and non-responder participants.

The serum levels of BDNF were significantly higher in patients who responded to the treatment (n = 21) compared to non-responders (n = 19): 29.98 ± 3.66 ng/mL vs. 20.16 ± 1.27 ng/mL (*p* < 0.0001, 95% CI 6.835 to 12.81). The effect size for the BDNF difference between groups was considerable (d = 4.66 [95% confidence interval −2.8, −1.3]) (Figure 1).

After comparing variables between responders and non-responders to treatment, we observed that previous depressive episodes (PDEs) were more common among patients who did not respond to antidepressants. Therefore, we included this variable as an independent variable in the regression model. According to the binary logistic regression (using response as a dependent variable and BDNF, previous depressive episodes, and their interaction as independent variables), we observed that BDNF was a good predictor of response (Odds ratio 2.28, 95% confidence interval 1.27–4.08), and the interaction between BDNF and the number of previous episodes was non-significant. Additionally, we observed an inverse relationship between pre-treatment BDNF serum levels and the severity of anxiety and depression symptoms after treatment, according to the final State and Trait Anxiety Inventory–state anxiety subscale (STAI-S) (r = −0.503, *p* = 0.001) and HDRS (r = −0.630, *p* < 0.001) scores.

Finally, the discriminating ability of basal BDNF serum levels to separate responders from non-responders was explored. The area under the curve (AUC) was 0.93, with a cut-off > 24.63 ng/mL (sensitivity = 0.929 and specificity = 0.857) (Figure 2).

## 3. Discussion

In our study, we analyzed the potential predictive role of BDNF in identifying responders and non-responders to antidepressants among MDD patients. We found that the serum concentrations of BDNF among MDD-diagnosed patients were different between responders and non-responders.

According to the neurotrophic hypothesis of depression [21,22], in which neurotrophic factors play a critical but not unique role in the pathogenesis of MDD, central and serum BDNF have been associated both with the presence of this diagnosis [23] and the effects of certain antidepressant treatments [24]. 

In our sample, higher basal BDNF serum levels were observed in patients who responded to antidepressants, opposite to the observation from Lopez and colleagues [25] who did not find differences in BDNF levels at the basal line, but after 8 weeks, differences between responders and non-responders were observed. It is well established that there is an interplay between BDNF and depressive disorder [26], especially as a good classifier of MDD diagnosis. The observation of low serum BDNF levels in depressed patients compared to healthy controls has been sustained by different studies [27,28,29]; therefore, we did not consider evaluating BDNF levels in healthy participants; however, data from healthy participants could add more value to our findings. Moreover, BDNF concentration increases after antidepressant treatment, especially with antidepressants that increase serotonin levels [28,30]. A possible explanation is that BDNF serves as a transducer, capable of positively regulating neuroplastic changes in neuronal populations that have undergone the neurobiological changes already described during MDD, especially in forebrain regions such as the hippocampus [31], resulting in an improvement in depressive symptoms. In this sense, the hypothesis could be considered that patients with a depressive syndrome could have a better response to antidepressant treatment if they have higher levels of BDNF in specific areas of the central nervous system, such as the hippocampus. Additionally, some characteristics of MDD could be associated with changes in BDNF such as the time elapsed with a non-treated depressive episode [12] or the number of previous episodes. In this sense, we observed no association between serum BDNF levels and the duration of the last episode and no differences between the responder and non-responder groups in the time elapsed since the onset of depressive symptomatology. Although we also did not observe a relationship between the number of PDEs and the level of BDNF, the patients who did not respond to treatment had more previous episodes than the responders. Considering this, we decided to include the number of PDEs in the logistic regression, which was not associated with response to treatment. This observation needs more exploration, considering that a relation between BDNF serum levels and BDNF gene mutations with recurrent depression has been consistently reported [32,33,34].

In line with other authors, i.e., Wolkowitz OM, et al. [35], we did not find a relationship between baseline BDNF levels and HDRS scores at entry, which were negatively correlated even in the first and second weeks of starting antidepressant treatment [36]. However, pre-treatment serum BDNF levels were inversely associated with the final HDRS and STAI-S scores in our patients, suggesting that BDNF could be a good predictor of response to treatment and that higher basal BDNF levels in depressed patients could indicate the relief of anxiety symptoms after an antidepressant assay. Some reports indicate a correlation between the severity of depression and BDNF levels [37], but the association with anxiety scores has been less explored. For example, a study reported that the anxiety trait was associated with BDNF polymorphism in patients with panic disorder [38]. Therefore, certain stress states could potentially influence the response to antidepressant treatment in this way. Even the association between peripheral BDNF and state anxiety could predict changes in somatic symptoms after treatment with escitalopram in patients with panic disorder [39]. Therefore, it would be valuable to investigate whether similar results can be found in depressed patients who have received treatment with this class of drugs.

BDNF levels, either alone or combined with insulin-like growth factor, demonstrated a good discriminating ability for MDD diagnosis [40]. Here, we added the value of discriminating responders and non-responders to treatment when levels are 24.63 ng/dl or higher. In this sense, one observed property of BDNF in clinical studies is its potential to predict antidepressant response, especially when it increases with the pharmacological assay. In contrast, it has been reported that in adolescents with MDD, responders showed a significant decrease in BDNF levels at week two of Selective Serotonin Reuptake Inhibitor (SSRI) treatment, compared to non-responders whose levels remained unchanged until that time [41]. The clinical usefulness of BDNF as a predictive factor in patients with treatment-refractory depression who have received nonconvulsive electrotherapy has even been denied due to the lack of changes at the serum level [42]. Considering these contradictory observations from previous studies, we believe that a limitation of our study was not reassessing BDNF levels after 8 weeks of antidepressant treatment. Clarifying inconsistencies between populations and age groups will give the potential predictive value to BDNF on an individualized basis. We consider that our results align with the potential of BDNF to act as a predictor of antidepressant response. However, considering the limited sample size of this study, it is important to note that our results only suggest that BDNF may have predictive significance. This needs to be proven in a larger cohort of participants. The impact of BDNF on anxiety symptoms should also be explored, considering its interference with antidepressant response.

Despite the limitations of our study, we believe that our results suggest important points for clinical practice, such as (1) the routine evaluation of anxiety in depressed patients, (2) the treatment of anxiety, independently of antidepressants, should be incorporated into the guides of treatment for depressive patients, and (3) special consideration should be given to patients with high levels of anxiety before or after treatment.

Lastly, we think about some perspectives in this field of research and consider that longitudinal studies with larger samples could help validate the function of pre-treatment BDNF levels in antidepressant response. These studies should include more types of antidepressants, both SSRIs and those from other families (SNRIs and atypical antidepressants, among others), and identify specific subgroups of patients for whom BDNF is a particularly strong predictor of antidepressant response, thereby enabling more personalized treatment strategies.

## 4. Materials and Methods

### 4.1. Participants

Participants were enrolled in the outpatient clinic and the first contact attention services at the National Institute of Psychiatry (INPRFM) in México City, between March 2022 and July 2023. 

Posters inviting participants were placed in the attention area with contact information for enrolling in this study. Eligible patients were those who met the diagnosis of major depressive disorder according to the 5th edition of the Diagnostic and Statistical Manual of Mental Disorders (DSM-5) [43]. The trial protocol was reviewed by the institutional ethics committee (CEI/C/010/2022), and all the participants were informed about methods and provided written informed consent before entering this study. The institutional scientific committee also approved the study protocol.

Inclusion criteria were defined as follows: depressed patients between 18 and 65 years of age, with at least 18 points scored on the 17-item HDRS (HDRS-17), and no pharmacological treatment for psychiatric conditions in the last 6 months. Exclusion criteria included psychotic or manic symptoms, substance use disorder according to DSM-5 criteria, psychiatric disorders other than depression, severe or uncontrolled medical illnesses, and hormonal supplements. 

### 4.2. Design

We conducted a prospective, open-label clinical study, which included three follow-up visits: (1) a basal evaluation at entry, (2) a second visit at week 4 to assess adherence to treatment, and (3) a final visit. A transversal measurement of basal BDNF was conducted, along with longitudinal evaluations of depressive and anxiety symptoms.

### 4.3. Evaluations

#### 4.3.1. Sociodemographic and Clinical Characteristics

Marital status was categorized as single, partnered, separated/divorced, and widowed. For education level, information was extracted on the highest completed educational level and was classified into four categories: elementary school, junior high school/high school, college, and other. Employment status was classified as unemployed, employed, student, or other. Lastly, religious affiliations were grouped into atheist, Catholic, Christian, and other.

We also assessed the number of previous depressive episodes, classified as 1:0 to 2 PDEs, 2:3 or 4 PDEs, and 3:5 or more PDEs, and the time since symptoms initiated in the last episode until the first evaluation in weeks (1:2 to 4 weeks, 2:5 to 8 weeks, 3:9 or more weeks), according to self-reports from the participants. The antidepressants prescribed were recorded at entry and at the final time, as well as the doses employed.

#### 4.3.2. AMAI 

The socioeconomic level of participants was evaluated using a tool developed by the Mexican Association of Market Research and Public Opinion Agencies, known as AMAI (Asociación Mexicana de Agencias de Inteligencia de Mercado). This tool is based on the development of a statistical model that objectively and quantifiably classifies households according to their socioeconomic status. This questionnaire measures the fulfillment of basic and key household needs, which include internet services, the number of bedrooms and bathrooms, the number of cars, and the number of employed individuals in the household, among other factors. AMAI categorizes economic status into seven distinct levels: A/B (high), C+ (upper middle), C (middle), C− (lower middle), D+ (low), D (poverty), and E (extreme poverty). This classification system is commonly used in Mexican research as it accurately reflects the key attributes of the population and ensures consistency in the classification of economic levels [44].

#### 4.3.3. HDRS

We used the 17-item version of the HDRS to evaluate depressive symptomatology and response to treatment. The HDRS is considered the gold standard for assessing the severity of depressive symptoms; it consists of 17 items rated on a scale of 0–4 (with 0 indicating the symptom is absent and 4 indicating it is severe); the total scores range from 0 to 54 [45,46]. An HDRS score of 7 or less post-treatment is commonly seen as an indicator of remission. A reduction of 50% or more from the baseline score during treatment is considered a sign of clinical response, indicating a clinically significant improvement. We used the validated Spanish version of the HDRS [45]. It was administered by an experienced psychiatrist and a trained physician.

#### 4.3.4. STAI-S 

The State–Trait Anxiety Inventory is a self-reported instrument widely used to evaluate state and trait anxiety in both the general and clinical populations. For this study, we used the state anxiety subscale, which evaluates the current state of anxiety by asking respondents how they feel “right now,” including items that measure subjective feelings of apprehension, tension, nervousness, worry, and the activation/arousal of the autonomic nervous system. It contains 20 items; higher scores indicate more anxiety [47]. We utilized the Spanish version of the STAI validated by Spielberger and Díaz-Guerrero [48]. The results demonstrated that this instrument possesses strong discriminative power in assessing anxiety and satisfactory validity and internal consistency. 

#### 4.3.5. Procedures

Basal line: Patients who were diagnosed with major depressive disorder according to the DSM-5 criteria and were prescribed an SSRI on the first visit to the outpatient clinic were evaluated before initiating treatment. The clinimetric instruments were applied at the initial visit, and a complete clinical interview was performed. We assessed sociodemographic variables, including socioeconomic level, marital status, occupation, and years of education. Previous depression episodes were documented according to the participants’ self-reports. The time from the initial symptoms until evaluation was recorded in weeks. At this time, we asked participants to complete the STAI-S. The evaluation included the HDRS application, conducted by an experienced clinician.

At the conclusion of the evaluation, the patients were scheduled for a lab appointment the next day under fasting conditions, and they attended their respective treating physician’s appointments, underscoring the pivotal role of the physician in the treatment process.

Medication was indicated by an independent psychiatrist according to their clinical judgement. Patients who were medicated with an SSRI were included in the following evaluations.

#### 4.3.6. Laboratory Test

Serum paraclinical tests were performed by the INPRFM clinical laboratory in the morning between 7 and 8 am with a 12 h fast, before initiating antidepressants. Lipid and partial thyroid profiles, fasting glucose, morning serum cortisol, and free and total testosterone were determined, and in the case of women, the gynecological hormonal profile was included. Human-free BDNF Quantikine ELISA Kits were used to assess serum BDNF concentrations in 96-well strip plates (R&D Systems, Inc. 614 McKinley Place NE; Minneapolis, MN 55413; USA). The global protein concentration in the centrifuged serum samples was determined by colorimetry using the Bradford test. Subsequently, dilutions were made at a concentration of 1:15 of the kit’s diluent reagent, according to the manufacturer’s standardized parameters, to read its optical density at 450 nm.

Week 4: On the second visit, we confirmed that the patients continued taking their medication, and if there was a change in dosage, we recorded that information. A blister count and questions about the effect of their medication were conducted.

Week 8: At the final evaluation, we again applied the clinimetric instruments: STAI-S and HDRS. Participants who had a reduction of more than 50% in their HDRS score were classified as responders to treatment (Figure 3).

#### 4.3.7. Statistical Analysis

Differences between groups were analyzed using a *t*-test or chi-squared test, as appropriate. Binary logistic regression was conducted using response to treatment as the dependent variable and BDNF, PDEs, and the BDNF/PDE interaction as independent variables. Pearson’s correlation test was conducted to assess the association between BDNF, STAI-S, and HDRS. Lastly, the discriminating ability of basal BDNF serum levels to separate responders from non-responders was explored by calculating the area under the receiver operating characteristic curve (AUC-ROC). A cut-off was derived from the ROC using Youden’s J criteria. SPSS version 21.0 was used for statistical analysis; GraphPad Prism 10 for macOS was used for generating graphs. The significance level was set at 0.05.

## 5. Conclusions

Pre-treatment BDNF levels may have predictive value for the antidepressant response in patients taking SSRIs; anxiety symptoms could remain elevated after antidepressant treatment in patients with low basal BDNF levels.

## Figures and Tables

**Figure 1 ijms-25-10373-f001:**
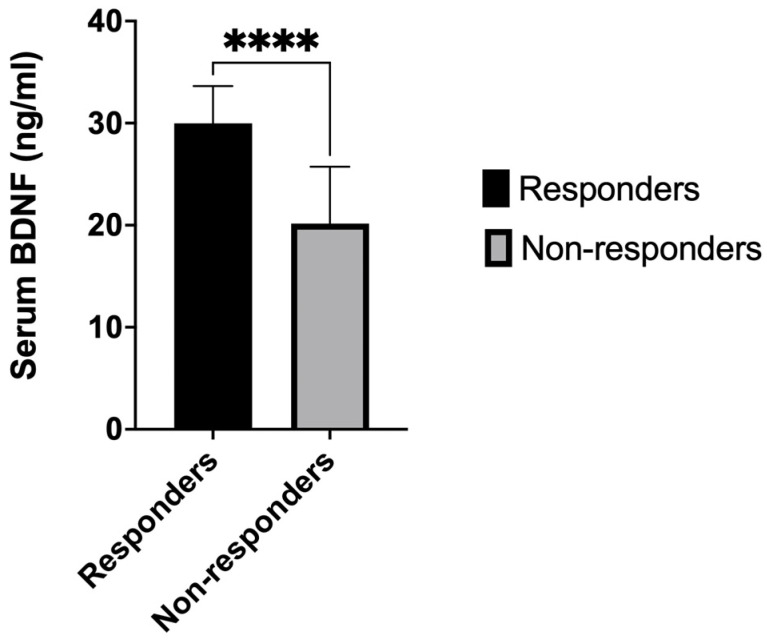
The quantification of BDNF in serum. Responder participants had higher pre-treatment baseline serum BDNF concentrations than non-responders. The results represent the mean ± the standard error of the mean. Asterisks indicate significance at *p* < 0.0001. The black square represents responders. The gray square represents non-responder patients.

**Figure 2 ijms-25-10373-f002:**
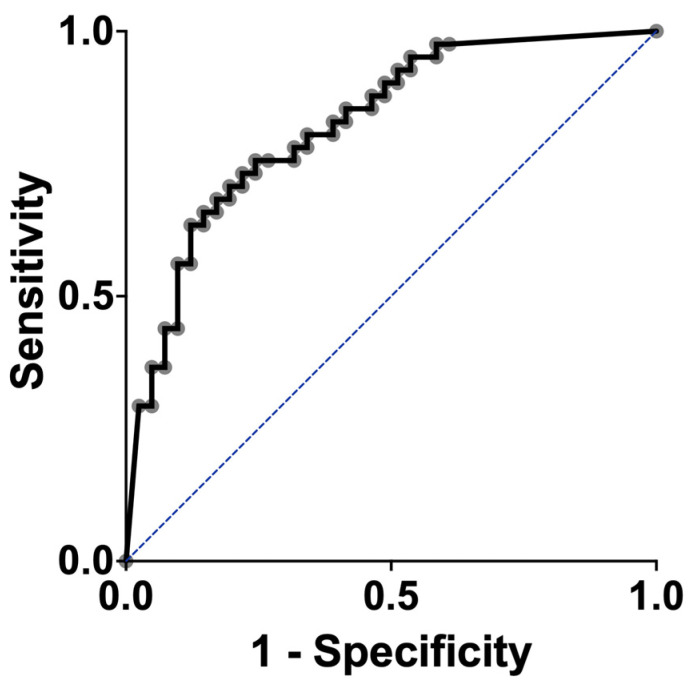
Serum BDNF as a predictor of the response to treatment in depression-diagnosed patients. The values of the ROC curve are shown (AUC= 0.9348, standard error = 0.03785, 95% CI = 0.8606 to 1.000, *p* < 0.0001, sensitivity = 0.929, and specificity = 0.857). AUC, the area under the ROC curve; BDNF, brain-derived neurotrophic factor. ROC, receiver operating characteristic).

**Figure 3 ijms-25-10373-f003:**
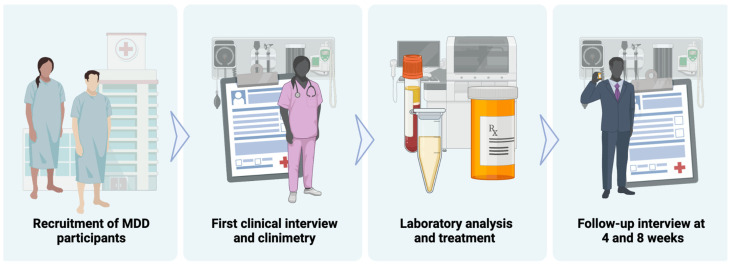
Timeline. Patients diagnosed with MDD and who had not started antidepressant treatment were recruited from the clinical area. Subsequently, the first clinical interview was conducted, and baseline clinimetric tests were applied. Clinical laboratory studies were requested, a serum sample was taken, and BDNF was measured by ELISA. An independent psychiatrist initiated pharmacological treatment. Follow-up interviews and clinimetric tests were carried out at 4 and 8 weeks. Abbreviations: MDD: major depressive disorder; BDNF: brain-derived neurotrophic factor; ELISA: Enzyme-Linked Immunoadsorption Assay. Illustrations created with BioRENDER (URL: https://www.biorender.com/ accessed on 6 August 2024).

**Table 1 ijms-25-10373-t001:** Demographic characteristics of responders and non-responders to antidepressants.

	Responders n = 21	Non-Responders n = 19	Statistic
Demographic Features			
Gender n (%)			χ^2^ = 0.93, df = 1, *p* = 0.45
Women	15 (71.4)	16 (84.2)	
Men	6 (28.6)	3 (15.8)	
Age (years) Mean S.D.	33.5 (11.71)	31.8 (11.27)	t = −4.75, df = 38, *p* = 0.63
Marital status n (%)			χ^2^ = 6.05, df = 3, *p* = 0.109
Single	13 (61.9)	15 (78.9)	
Partnered	6 (28.6)	2 (10.5)	
Separated/divorced	0 (0.0)	2 (10.5)	
Widow	2 (9.5)	0 (0.0)	
Education n (%)			χ^2^ = 1.03, df = 3, *p* = 0.793
Elementary school	1 (4.8)	1 (5.3)	
High school	9 (42.9)	6 (31.6)	
College	9 (42.9)	11 (57.9)	
Graduate school	2 (9.5)	1 (5.3)	
Occupation n (%)			χ^2^ = 3.34, df = 3, *p* = 0.342
Unemployed	6 (28.6)	4 (21.1)	
Employed	6 (28.6)	5 (26.3)	
Students	5 (23.8)	9 (47.4)	
Other	4 (19)	1 (5.3)	
Religion n (%)			χ^2^ = 7.116, df = 3, *p* = 0.068
Atheist	6 (28.6)	8 (42.1)	
Catholic	13 (61.9)	6 (31.6)	
Christian	0 (0.0)	4 (21.1)	
Other	2 (9.5)	1 (5.3)	

**Table 2 ijms-25-10373-t002:** The clinical characteristics of the participants.

	Responders n = 21	Non-Responders n = 19	Statistic
Clinical Characteristics			
	Mean (S.D.)	Mean (S.D.)	
Basal HDRS	23.04 (4.32)	22.26 (4.4)	t = −0.56, df 38, *p* = 0.28
Basal STAI-S	50.30 (16.96)	48.94 (19.23)	t = −0.23, df 38, *p* = 0.81
BDNF	29.98 (3.66)	20.16 (2.27)	t = −6.65, df 38, *p* = 0.0001
BMI	24.35 (4.13)	27.24 (7.47)	t = −1.50, df 38, *p* = 0.14
LDE n (%)			
1.00	7 (58.3)	5 (41.7)	χ^2^ = 1.27, df 2, *p* = 0.52
2.00	0 (0.0)	1 (100)	
3.00	14 (51.9)	13 (48.1)	
PDE n (%)			
1.00	12 (57.1)	4 (21.1)	χ^2^ = 6.16, df 2, *p* = 0.046
2.00	2 (9.5)	6 (31.6)	
3.00	7 (33.3)	9 (47.4)	
Antidepressant n (%)			χ^2^ = 0.57, df 3, *p* = 0.902
Citalopram	1 (4.8)	2 (10.5)	
Escitalopram	4 (19.0)	3 (15.8)	
Fluoxetine	11 (52.4)	9 (47.4)	
Sertraline	5 (23.8)	5 (26.3)	

HDRS: Hamilton Depression Rating Scale, STAI-S: State Anxiety Inventory, BDNF: brain-derived neurotrophic factor. BMI: Body mass index, LDE: last depressive episode duration in weeks (1: 2 to 4 weeks, 2: 5 to 8 weeks, 3: 9 or more weeks): 1, PDE: previous depressive episode (1: 0 to 2 PDEs, 2: 3 or 4 PDEs, and 3: 5 or more PDEs).

## Data Availability

Data are unavailable because the participants did not agree to publish their complete information; they only accepted the publication of the study results without data that could identify them.

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
