# Peer review of "Are BDNF and Stress Levels Related to Antidepressant Response?"

_ijms, 2024, doi:10.3390/ijms251910373_

Round 1

Reviewer 1 Report

Comments and Suggestions for Authors

IJMS

Title: "Are BDNF and Stress Levels Related to Antidepressant Response? by Mónica Flores-Ramos et al. reveals that baseline BDNF levels may predict antidepressant response to SSRIs, with higher levels found in treatment responders. Lower BDNF levels were associated with increased post-treatment anxiety. However, more research is needed to determine the specific effect of stress in therapy responses.

The author stated that rigorous approach, detailed analysis, and clear presentation of findings provide a significant addition to the discipline. Their findings not only advance our understanding of BDNF function in brain health, but also lay the groundwork for future research and therapeutic applications aimed at improving outcomes for those suffering from neurological illnesses.

The manuscript should be considered for publication in IJMS by addressing the few points given below.

Please read the manuscript thoroughly and correct any seemingly insignificant grammatical errors.

1.      There are multiple grammatical errors in the paper, as well as issues in capitalization, and sentence structure, including missing punctuation and conjunctions.

Line number 42, response rate reported Is 47%, according to the Sequenced treatment alternatives to relieve 42 depression (STAR*D) [2]

Add a period after the citation, 2, 3, 4-6, 7, 11, 12, etc.

Line no. 66, being lower the BDNF levels in female patients than female controls, please correct the sentence.

2.      At the end of the introduction, clearly describe the study's main goal and hypothesis.

For example:  the current study was conducted to determine if plasma BDNF levels differ between depressive individuals who react to antidepressant medication and those who do not respond, as well as the impact of anxiety in this condition.

3.      What criteria were used to choose the 40 depressed adults for the study? Were there any inclusion or exclusion criteria?

4.      How were the STAI and HDRS administered? Were they self-reported or administered by a clinician?

5.      When were BDNF levels and anxiety measured in relation to the 8-week treatment period?

6.      Can you address on the suggested processes by which BDNF regulates depression and anxiety? How could these mechanisms interact?

7.      How might the findings of this study influence clinical practice or the development of novel treatments for MDD with concomitant anxiety?

8.      Can you explain how the binary logistic regression model was defined, including any interaction terms or covariates?

9.      Conclusion: Could more longitudinal studies help to validate the predictive function of pre-treatment BDNF levels in antidepressant response, especially across different SSRIs and patient populations? Please elaborate a little more.

Comments on the Quality of English Language

 There are multiple grammatical errors in the paper, as well as issues in capitalization, and sentence structure, including missing punctuation and conjunctions.

Author Response

Thank you very much for taking the time to review this manuscript. Please find the detailed responses below and the corresponding revisions/corrections highlighted/in track changes in the re-submitted files. We consider that your comments are important to improve our manuscript.

3. Point-by-point response to Comments and Suggestions for Authors

Comments 1: There are multiple grammatical errors in the paper, as well as issues in capitalization, and sentence structure, including missing punctuation and conjunctions.

Response 1:  Sorry for this inconvenience, re reviewed the grammatical errors in the paper.

Line number 42, response rate reported Is 47%, according to the Sequenced treatment alternatives to relieve 42 depression (STAR*D) [2]

We corrected the sentence:

Currently, many pharmacological options are available to treat depression; and despite their efficacy, the response rate reported is 47%, according to the Sequenced Treatment Alternatives to Relieve Depression (STAR*D) study [2].

Add a period after the citation, 2, 3, 4-6, 7, 11, 12, etc.

Sorry for the inconvenience, we added the period as corresponded.

Line no. 66, being lower the BDNF levels in female patients than female controls, please correct the sentence.

Thank you for the observation; we changed the sentence to: “serum BDNF levels did not differ from those of healthy controls, but an association with gender was observed: BDNF levels were lower in female patients than in female controls” (page 2, lines 66-68).

Comments 2:  At the end of the introduction, clearly describe the study's main goal and hypothesis.

For example:  the current study was conducted to determine if plasma BDNF levels differ between depressive individuals who react to antidepressant medication and those who do not respond, as well as the impact of anxiety in this condition.

Response 2: Thank you for your suggestion, the objective of the study was changed to:

Therefore, the current study was conducted to determine if plasma BDNF levels differ between depressed patients who respond to antidepressant treatment and those who do not respond, as well as the impact of anxiety in this condition (page 2, lines 75-78).

Comment 3: What criteria were used to choose the 40 depressed adults for the study? Were there any inclusion or exclusion criteria?

Response 3:

Yes, we established some criteria for inclusion or exclusion, and are reported in the section 4.1 Participants, as follows:

Inclusion criteria were defined as follows: depressed patients between 18 to 65 years of age, at least 18 points scored in the 17 items Hamilton Depression Rating Scale (HDRS-17), and no pharmacological treatment for psychiatric conditions in the last 6 months. Exclusion criteria included psychotic or manic symptoms, substance use disorder according to DSM-5 criteria, psychiatric disorders other than depression, severe or uncontrolled medical illnesses, and hormonal supplements.

Comment 4: How were the STAI and HDRS administered? Were they self-reported or administered by a clinician?

Response 4: STAI was self-reported, and HDRS was administered by an experimented Psychiatrist and a trained physician, members of the research team. We added this information at the Evaluations section, as follows:

It was administered by an experimented psychiatrist and a trained physician.

Comment 5: When were BDNF levels and anxiety measured in relation to the 8-week treatment period?

Response 5: The basal evaluation (including STAI, HDRS) were done in the first visit, and the next day was the laboratory test in fasting condition. We had to take the blood sample in the next day before patients began treatment. Therefore, no more than a day passed between the evaluations and the sample collection. After the basal evaluation, 8 weeks of treatment started.

We add this sentence in the Procedures section: At the conclusion of the evaluation, the patients were scheduled for a lab appointment the next day in fasting conditions (page 9, lines 271-272).

Comment 6: Can you address on the suggested processes by which BDNF regulates depression and anxiety? How could these mechanisms interact?

Response 6: Anxiety and depression share a common neurobiological basis. In animal studies, acute or chronic stress protocols are used to induce depression-like behavior. It is known that the HPA axis plays a very important modulatory role through the secretion of corticosteroids. Additionally, stress leads to a decrease in BDNF levels, which is essential for the neurogenesis process and a good response to antidepressant treatment. Some scientific studies have reported that antidepressant drugs can restore BDNF levels. Clinical studies have observed that up to 60% of depressed patients present with anxiety symptoms.

In the context of depression, reduced levels of BDNF, particularly in the hippocampus and prefrontal cortex, have been associated with the onset and persistence of depressive symptoms. Antidepressant treatments often work, at least in part, by increasing BDNF levels, which helps restore normal neuroplasticity and synaptic function.

In anxiety, BDNF also appears to be a key player. Animal models have shown that BDNF signaling in the amygdala is important for the formation and extinction of fear memories, which are central to anxiety disorders. Alterations in BDNF levels and signaling in the amygdala may contribute to the heightened fear response and difficulty in fear extinction seen in anxiety disorders.

Suggested mechanisms that explain this interrelationship include neurogenesis process, synaptic plasticity and stress response modulation. These mechanisms could be interacting to perpetuate the presence of anxious and depressive symptoms in certain patients, in a bidirectional influence or a synergic mechanism.

Comment 7: How might the findings of this study influence clinical practice or the development of novel treatments for MDD with concomitant anxiety?

Response 7: We consider that the findings of our study could be clinically relevant in the treatment of depression in several ways: 1) in the clinical practice evaluation of anxiety level should be rutinary, 2) patients with high levels of anxiety after an antidepressant essay probably need another kind of antidepressant, different to SSRI; 3) patients without response to antidepressant treatment could have another interventions that potentially increase BDNF levels, such as exercise, diet interventions, neuromodulation therapy or ketamine treatment; 4) Treatment of anxiety, independently of antidepressant, should be incorporated into the guides of treatment for depressive patients.

According to your comment, we added a small paragraph in the discussion section:

Despite the limitations of our study, we believe that our results suggest important points for clinical practice, such as 1) routine evaluation of anxiety in depressed patients, 2) treatment of anxiety, independently of antidepressant, should be incorporated into the guides of treatment for depressive patients, 3) special consideration should be given to patients with high levels of anxiety before or after treatment (pages 7-8, lines 201-205).

Comment 8: Can you explain how the binary logistic regression model was defined, including any interaction terms or covariates?

Response 8: Initially, we compared variables that might influence BDNF levels between patients who responded to treatment and those who did not (including gender, age, BMI, last depressive episode duration and PDE). The variable that showed a significant difference between groups (previous depressive episodes) was included in the logistic regression model, along with its interaction with BDNF, to assess the potential confounding effect of number of previous depressive episodes. Therefore, dependent variable was response to treatment (yes or not) and predictors were BDNF, previous depressive episodes, and their interaction.

We added the next brief explanation in the manuscript:

After comparing variables between responders and non-responders to treatment, we observed that PDE were more common among patients who did not respond to antidepressants. Therefore, we included this variable as an independent variable in the regression model (page 3, lines 103-106).

Comment 9: Conclusion: Could more longitudinal studies help to validate the predictive function of pre-treatment BDNF levels in antidepressant response, especially across different SSRIs and patient populations? Please elaborate a little more.

Response 9: You are right, we consider that longitudinal studies with larger samples could help to validate the function of pre-treatment BDNF levels in antidepressant response. It is convenient to enhance the understanding of antidepressant response mechanisms. These studies should particularly include more types of antidepressants, both SSRIs and those from other families (SNRIs, atypical antidepressants, among others), as it has been clinically observed that some are more anxiolytic than others, and in certain types of patients, some antidepressants work better than others. BDNF levels in a basal study time, may reflect many other conditions, that we are unaware of; however, to determine this, a more in-depth evaluation of the patient's prior history is required. This approach could help to identify specific subgroups of patients for whom BDNF is a particularly strong predictor of antidepressant response, thereby enabling more personalized treatment strategies.

According to your comment we added a text in the conclusion:

Lastly, we think about some perspectives in this field of research and consider that longitudinal studies with larger samples could help to validate the function of pre-treatment BDNF levels in antidepressant response, including include more types of antidepressants, both SSRIs and those from other families (SNRIs, atypical antidepressants, among others), and identifying specific subgroups of patients for whom BDNF is a particularly strong predictor of antidepressant response, thereby enabling more personalized treatment strategies (page 8, lines 206-212) .

Reviewer 2 Report

Comments and Suggestions for Authors

The study aimed to analyze how BDNF impacts antidepressant response, considering anxiety levels. The manuscript presents the clinical data and characteristics of patients enrolled in the study, with serum BDNF levels assessed before starting SSRI treatment. The authors concluded that the baseline level of serum BDNF could be a promising predictor of patients' response to SSRIs, potentially offering a significant contribution to the field.

However, severe issues with the study design are detected, lowering the presented data's scientific soundness. Moreover, other authors had already published the same conclusions much earlier.

Detailed comments:

- First, there are no results from healthy patients to show control BDNF levels and perform appropriate comparisons.

- Second, based on the study design described in the methodology section, the serum level of BDNF was assessed only at a one-time point (before SSRI treatment), which does not allow the authors to define the presented results as a valid longitudinal study. A longitudinal study (detailed in Chapter 4.2), would provide a more comprehensive understanding of the relationship between BDNF and SSRI response, enhancing the scientific rigor of the study.

- Third, there is no information on which specific SSRI medication was used in the patients, or whether only one type of antidepressant was administered or different medicines were used.

- Fourth and most importantly, the presented results do not contribute anything new to the topic of BDNF and depression/antidepressants. A similar experimental design was already used by Wolkowitz et al. 2011 [Wolkowitz OM, Wolf J, Shelly W, Rosser R, Burke HM, Lerner GK, Reus VI, Nelson JC, Epel ES, Mellon SH. Serum BDNF levels before treatment predict SSRI response in depression. Prog Neuropsychopharmacol Biol Psychiatry. 2011 Aug 15;35(7):1623-30. doi: 10.1016/j.pnpbp.2011.06.013.], the conclusions were the same as those presented in the current study.

Author Response

1. Summary

Thank you very much for taking the time to review this manuscript. Please find the detailed responses below and the corresponding revisions/corrections highlighted/in track changes in the re-submitted files. We consider that your comments are important to improve our manuscript.

3. Point-by-point response to Comments and Suggestions for Authors

Comments 1: There are no results from healthy patients to show control BDNF levels and perform appropriate comparisons.

Response 1: Thank you for your valuable feedback; we understand the importance of including healthy controls in a clinical evaluation. However, our main objective was to investigate the difference in BDNF pretreatment levels in depressed patients who responded to antidepressant treatment and those who did not. In our knowledge, many scientific reports agree that BDNF levels differ between depressed patients and healthy controls; that’s why we considered not to be necessary to include controls in our study; additionally, they had not the intervention (antidepressant). But, thanks to your observation, we have added this observation in the discussion section of the manuscript, as follows:

“Therefore, we did not consider evaluating BDNF levels in healthy participants; however, data from healthy participants could add more value to our findings (page 6, lines 145-147).

Comment 2: Based on the study design described in the methodology section, the serum level of BDNF was assessed only at a one-time point (before SSRI treatment), which does not allow the authors to define the presented results as a valid longitudinal study. A longitudinal study (detailed in Chapter 4.2), would provide a more comprehensive understanding of the relationship between BDNF and SSRI response, enhancing the scientific rigor of the study.

Response 2: I greatly appreciate your feedback; we agree with your comment considering the BDNF evaluation; however, the repeated observations (an essential characteristic of the longitudinal studies) were done in the clinimetric assessment. According to your suggestion we make clear this consideration in the Chapter 4.2 as follows:

We conducted a prospective, open-label clinical study, which included three follow-up visits: 1) Basal evaluation at entry, 2) a second visit at week 4 to assess adherence to treatment, and 3) a final visit. A transversal measurement of basal BDNF was con-ducted, and longitudinal evaluations of depressive and anxiety symptoms.

Moreover, we added the next sentence at page 7, lines 193-195:

Considering those contradictory observations from previous studies, we believe that a limitation of our study was not reassessing BDNF levels after 8 weeks of antidepressant treatment.

Comment 3: There is no information on which specific SSRI medication was used in the patients, or whether only one type of antidepressant was administered or different medicines were used.

Response 3: According to your valuable comment we added in the text the description of the antidepressants used for the participants, and the distribution of them according to response or not response to treatment (no significant differences were observed between groups considering the prescribed antidepressant). We added the next information at the end of table 2:

Antidepressant n(%)

Citalopram

Escitalopram

Fluoxetine

Sertraline

1 (4.8)

4 (19.0)

11 (52.4)

5 (23.8)

2 (10.5)

3 (15.8)

9 (47.4)

5 (26.3)

χ2=0.57 df 3, p=0.902

Additionally, we reported in the results section (page 2, lines 91-93) that “No differences were observed in the antidepressants prescribed, between responder and non-responder participants”.

Comment 4: The presented results do not contribute anything new to the topic of BDNF and depression/antidepressants. A similar experimental design was already used by Wolkowitz et al. 2011 [Wolkowitz OM, Wolf J, Shelly W, Rosser R, Burke HM, Lerner GK, Reus VI, Nelson JC, Epel ES, Mellon SH. Serum BDNF levels before treatment predict SSRI response in depression. Prog Neuropsychopharmacol Biol Psychiatry. 2011 Aug 15;35(7):1623-30. doi: 10.1016/j.pnpbp.2011.06.013.], the conclusions were the same as those presented in the current study.

Response 4: Thank you for the observation, we previously read and cited the mentioned article. As you commented a similar design was used for the authors and conclusions are like ours. However, we added some important elements that are relevant from a theoretical perspective; namely, we evaluated the participants' anxiety levels, which, according to animal and clinical studies, may affect BDNF levels. Additionally, it is suggested that the number of previous depressive episodes is important for the neurobiological changes in depression such as BDNF levels and neurogenesis process, affecting the antidepressant response. Therefore, we assessed in our participants the number of previous depressed episodes. In this sense, we found that there was a difference in the number of PDEs between patients who responded to the treatment and those who did not, and therefore we included this variable in the statistical analysis. Finally, we provided a BDNF value (as a cutoff point > 24.63 ng/ml, with a good level of specificity and sensitivity) that allows for distinguishing between patients who will respond to the treatment and those who will not, which could be used in future clinical studies.

Round 2

Reviewer 1 Report

Comments and Suggestions for Authors

I am delighted to notify you that your work, "Are BDNF and Stress Levels Related to Antidepressant Response?" can be accepted in its current form. We value the thorough work you did in responding to all of the reviewers' comments and suggestions. Your comprehensive and thoughtful modifications considerably improved the manuscript's quality.

Thank you for your great contribution to this field. We look forward to seeing your work published and working with you again in the future.

Author Response

Thank you very much for your thorough review and positive feedback. We are pleased that you find our manuscript ready for publication in its current form. We appreciate your valuable input throughout the review process.

We corrected the abbreviations as suggested for the other reviewer.

Reviewer 2 Report

Comments and Suggestions for Authors

The authors have addressed mainly my comments and revised the manuscript according to the recommendations.

However, I have an additional comment: since the cohort of patients in the studied groups is small and significantly too limited to draw general conclusions regarding the use of the results in diagnostics, the authors should clearly emphasize this in the conclusions. Please indicate that "the results only suggest that BDNF measurement may have predictive significance."

Additionally, a minor editorial note: please ensure consistency in using abbreviations throughout the manuscript. The complete form of an abbreviation should be provided at its first occurrence.

Author Response

We truly appreciate your comments and believe that the changes you suggest are of great importance for a scientific report. We have reviewed the abbreviations, and additionally, in the discussion and conclusion, we have emphasized that BDNF may have predictive value. The changes are marked in bold in the text.

Thank you again for your suggestions